# Antimicrobial-resistance of *Escherichia coli* in dogs and cats: A scoping review

Rasaq A. Ojasanya[1]*, J. Scott Weese[2,3], Zvonimir Poljak[1], Kurtis E. Sobkowich[1], Uththami Kukathasan[1], Theresa M. Bernardo[1]

**1** Department of Population Medicine, Ontario Veterinary College, University of Guelph, Ontario, Canada,
**2** Department of Pathobiology, Ontario Veterinary College, University of Guelph, Ontario, Canada,
**3** Centre for Public Health and Zoonoses, Ontario Veterinary College, University of Guelph, Ontario, Canada

* ojasanyr@uoguelph.ca

## Abstract

Pathogenic *Escherichia coli* causes a range of clinical manifestations in dogs and cats, and the use of antimicrobials in pets is associated with the risk of antimicrobial resistance (AMR). Pets contribute to the dissemination of AMR both within their species  and to humans. This study conducts a scoping review to assess the existing evidence on the AMR of *E. coli* in dogs and cats, noting the purpose of antimicrobial susceptibility testing (AST) and determining the knowledge gaps to inform future research. The search utilized specific and generic strings aligned with the research objectives, spanning databases such as MEDLINE®, Web of Science, Biological Science Collection, AGRICOLA, CAB Direct, and Google Scholar, from January 1990 to July 2023. The study selection included only articles published in English and related to primary research. Following deduplication, the initial search identified 1,205 studies. After a detailed full-text review, 108 independent studies were identified. Studies on the AMR of *E. coli* in companion animals are largely concentrated in North America and Western Europe. Most of the studies were observational and were conducted in veterinary clinics. AST was primarily conducted to guide the antimicrobial treatment of *E. coli* infections in pets. Although not all studies provided clinical histories, among those that did, multi-drug resistant (MDR) *E. coli* was reported in both healthy and ailing pets. The detection of MDR *E. coli* in healthy and sick pets serve as a clarion call for antimicrobial stewardship. However, the limited number of studies dedicated to AMR monitoring and surveillance programs for companion animals raises a substantial concern.

## Introduction

In companion animals, pathogenic *Escherichia coli* can cause gastroenteritis or transition to extraintestinal sites, leading to urinary tract infections (UTIs), pyometra, and

**Data availability statement:** Relevant data are within the manuscript and its Supporting Information files.

**Funding:** The authors would like to thank IDEXX Laboratories for their support of the IDEXX Chair in Emerging Technologies and Preventive Healthcare, which funded this research. The funders had no role in study design, data collection and analysis, decision to publish, or preparation of the manuscript.

**Competing interests:** NO authors have competing interests

in severe cases, systemic infections [1]. The use of antimicrobials to treat bacterial infections in companion animals, particularly without prior antimicrobial susceptibility examination, has been identified as a significant contributor to the development of antimicrobial resistance (AMR) [2,3].

Companion animals, such as dogs and cats, are increasingly affected by AMR, posing a threat to their health and potentially to humanhealth [3,4]. Multi-drug resistant (MDR) *E. coli* has been identified in both clinically healthy and sick pets [5,6]. The emergence of extended-spectrum β-lactamase-producing *E. coli* (ESBL-*E. coli*) is a rising global threat among humans and companion animals [7,8]. Several investigations have hypothesized that extraintestinal pathogenic *E. coli* (ExPEC) possesses zoonotic potential, especially ExPEC isolates recovered from UTIs in dogs and cats [9–11]. Companion animals have the potential to serve as reservoirs of AMR due to their exposure to antimicrobials and their close association with humans [12]. Limited surveillance efforts and insufficient understanding of the extent of AMR in companion animals have been identified as critical challenges [3]. *E. coli* is particularly significant among bacterial pathogens due to its ubiquity, its capacity to acquire and disseminate AMR genes, and its role as an indicator organism for monitoring AMR trends.

This study aims to conduct a scoping review to identify and summarize existing evidence on the AMR of *E. coli* in dogs and cats, highlight the purpose of antimicrobial susceptibility testing, and identify knowledge gaps to inform future research. A scoping review was chosen as its aligns with the broad objectives of this research, allowing for an exploration of the extent and nature of existing studies and providing a descriptive overview of the research question [13].

## Materials and methods

### Protocol and registration

A research protocol for this study was developed by RO, reviewed and revised by knowledge experts (JW, ZP, KS, and TB), and published prior to the initiation of the scoping review. The protocol can be accessed at the University of Guelph's Atrium: https://hdl.handle.net/10214/27951 (accessed on February 15, 2024). The reporting of this protocol follows the Preferred Reporting Items for Systematic Reviews and Meta-Analyses (PRISMA) protocol guidelines [14], with the checklist provided in S1 Checklist. Details of any deviations from the protocol, along with justifications, are outlined in S1 Appendix A.

The reporting of this scoping review adheres to the PRISMA Extension for Scoping Reviews guidelines [15], with the checklist available in S2 Checklist.

### Research question

"What is the current global state of evidence regarding the AMR of *E. coli* in dogs and cats, the purpose of antimicrobial susceptibility testing (AST), and the existing knowledge gaps to inform future research?"

**Eligibility criteria.** To qualify for inclusion in this scoping review, a research study had to meet the following criteria:

1. The article must be a full-text publication in English, published from 1990 onwards.

2. The study must involve primary research (including observational studies such as cross-sectional, case-control, cohort, etc., or experimental studies such as randomized controlled trials (RCTs)). Other study types were quantified during the primary screening (title and abstract screening) but were excluded from the secondary screening (full-text screening).

3. The study must address the AMR of *E. coli* in dogs and/ or cats.

**Information sources.** The search encompassed databases such as MEDLINE® via Ovid, Web of Science (Core Collection), Biological Science Collection via ProQuest, AGRICOLA, and CAB Direct (CABI). Additionally, a search for gray literature was conducted using Google Scholar. The search was limited to articles published from January 01, 1990, onwards.

**Search strategy.** Our search query included keywords such as "antimicrob*", "anti-microb*", "antibiot*", "anti-biot*", "antibacter*", "anti-bacter*", "anti-bacterial agents", "drug resistance", "microbial", "resistance", "susceptibility", and "susceptible"; as well as "Escherichia", "Enterobacteriaceae", "Enterobacterales", and "E. coli"; and "companion animals", "pets", "dogs", "canine", "cats", and "feline". Boolean operators such as "AND" and "OR" were used to structure the query effectively. The search was restricted to English-language publications and applied across the six databases from January 1, 1990, to July 9, 2023.

Publications identified through various information sources were organized in Zotero® 6.0.26, and deduplication was carried out using the internal algorithms of this reference manager. The deduplicated citations were then imported into DistillerSR® (Copyright © 2008–2022, Evidence Partners Inc.), a review management software, where additional deduplication procedures were conducted. A final manual deduplication was performed on all references included in the data charting after the full-text screening.

**Selection of sources of evidence.** In DistillerSR®, study selection was carried out by two independent reviewers who applied eligibility criteria to all citations. Conflicts were resolved by a third reviewer at the end of each stage of the screening and the data charting process. The primary screening, secondary screening, and data charting forms were pre-tested on 50 articles in DistillerSR®, and the forms were adjusted as needed. Both screenings and data charting were conducted independently by two reviewers, with agreement determined at the answer level (either to include or exclude).

The primary screening process involved evaluating primary research articles based on predetermined questions, including criteria such as publication date, language, relevance to domestic dogs and/or cats, mention of *E. coli*, AST of isolated *E. coli*, and the study design (e.g., observational study, ongoing monitoring and surveillance programs, RCTs). The reviewers considered a study to have adopted an observational study design if it was cross-sectional, case-control, or cohort. Studies aiming to understand the molecular basis of resistance and laboratory studies for the validation of AST methods were excluded. Descriptive studies, such as case reports and case series, that were pertinent to the research question were categorized as "relevant but of secondary priority" and were ultimately excluded from this review. If the study type was unclear, such publications were advanced to the secondary screening stage.

Full texts for citations in the secondary screening were obtained through the University of Guelph's electronic library resources and relevant assessments were conducted for the retrieved full-text articles using the screening questions above (modified for the secondary screening).

**Data charting.** All full-text articles that advanced beyond the secondary screening phase were transferred to the data charting section in DistillerSR®. Comprehensive information was extracted from each full-text article, including details from the title, abstract, materials and methods, and results sections. The data items collected from the articles are summarized in S2 Appendix B. All retrieved articles, including those excluded with reasons for exclusion, and the data charted from the eligible studies, are summarized in S1 Table.

## Data analysis

The level of agreement between the two independent reviewers was assessed at the end of each stage of the study selection process - before conflict resolution - using the Kappa coefficient in DistillerSR®. A Kappa coefficient greater than 0.80 indicated a high level of agreement [16]. The level of agreement was re-assessed after conflict resolution at the end of each stage of the screening process to ensure consistency in the review process. A PRISMA flowchart was generated using DistillerSR® to illustrate the number of studies retrieved from the database search and the subsequent study selection processes. The extracted data were then summarized and visualized using Stata (StataCorp, College Station, TX, USA, 2015). The shapefile of the world was retrieved from the Database of Global Administrative Areas [17], imported into R (Core Team, 2023), and used to represent the distribution of retrieved publications on the map.

## Results

The search identified 1,522 citations from six electronic databases (published from January 1990 to July 2023). Duringde-duplication using DistillerSR®, 317 duplicates were excluded. A total of 1,205 sources underwent primary screening, and 1,052 records were further excluded because they were "relevant but of secondary priority" studies. These included studies to understand the molecular basis of resistance, AMR validation studies, non-primary research (e.g., narrative reviews, scoping reviews, meta-analyses, editorials, and opinion pieces), and studies that did not specifically address AMR of *E. coli* in dogs or cats. At full-text eligibility screening of 153 articles, 45 records were further excluded because they were descriptive studies, non-primary research, duplicates, or studies that did not address AMR of *E. coli* in dogs or cats. A total of 108 independent studies were included in the final data charting stage (Fig 1).

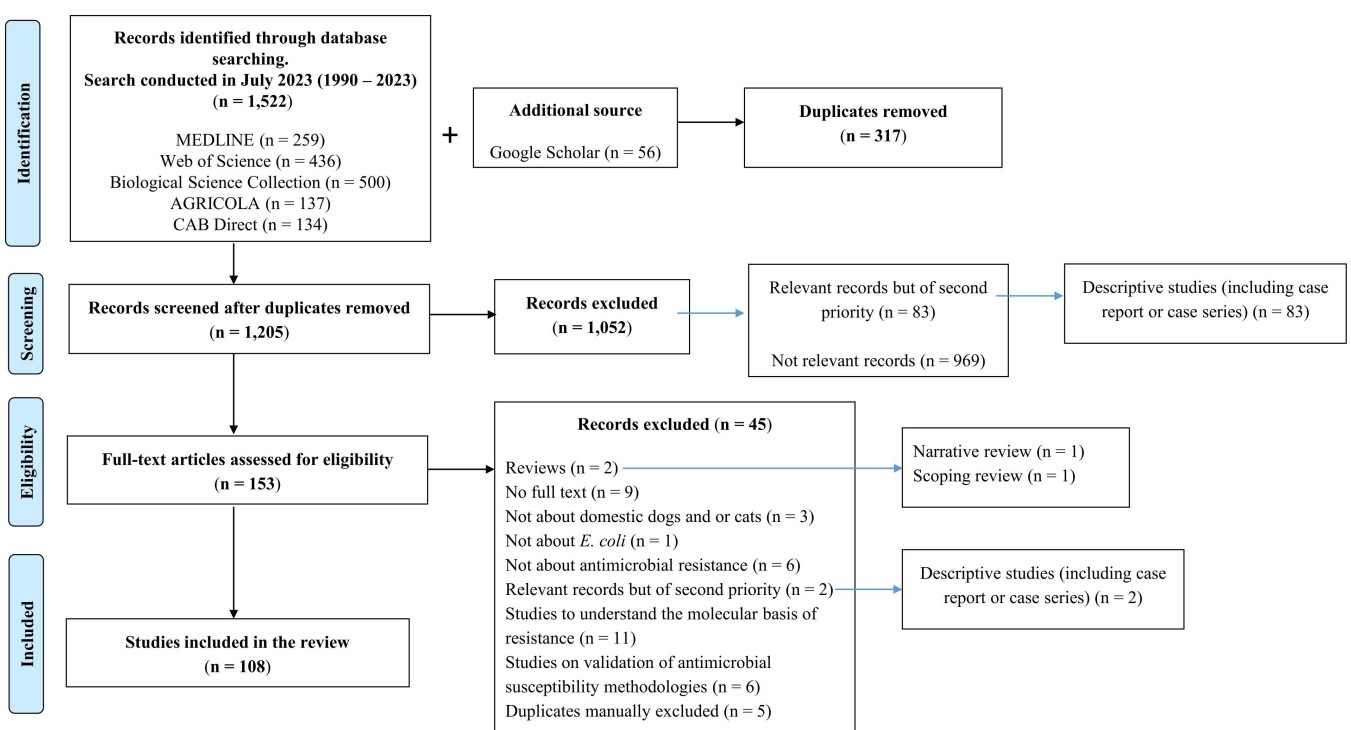

**Fig 1. The PRISMA flowchart shows the literature search and the study selection process for the antimicrobial resistance of *E. coli* in dogs and/or cats.** See S2 Checklist.

All relevant information that addressed our research questions, as outlined in the protocol, was collected at the data charting stage. The reviewers encountered conflicts regarding 24 studies at the end of the data charting stage, and the Kappa coefficient was 0.75 (substantial agreement). The Kappa coefficient later increased to 0.98 after the conflicts were resolved with the assistance of a third reviewer. The number of conflicts encountered and the Kappa coefficient before and after conflict resolution at each stage of the study selection process are summarized in S2 Table.

Predominant study locations were the United States (23.1%, n = 25), Italy (10.2%, n = 11), Germany (10.2%, n = 11), the United Kingdom (10.2%, n = 11), and Canada (9.3%, n = 10) (Fig 2). Studies conducted in multiple countries were counted individually.

Since 1990, there has been an apparent increase in the number of studies undertaken to investigate the AMR of *E. coli* in dogs and/or cats (Fig 3). The mean study duration was three years, as shown by the red reference line in Fig 4. However, only 28.7% (n = 31) of the studies had a duration of three years or longer, while the majority (58.3%, n = 63) were conducted in under three years. Additionally, some studies (13.0%, n = 14) did not provide specific details regarding their study duration.

The most commonly affiliated institutions of the first author are presented in S3 Table.

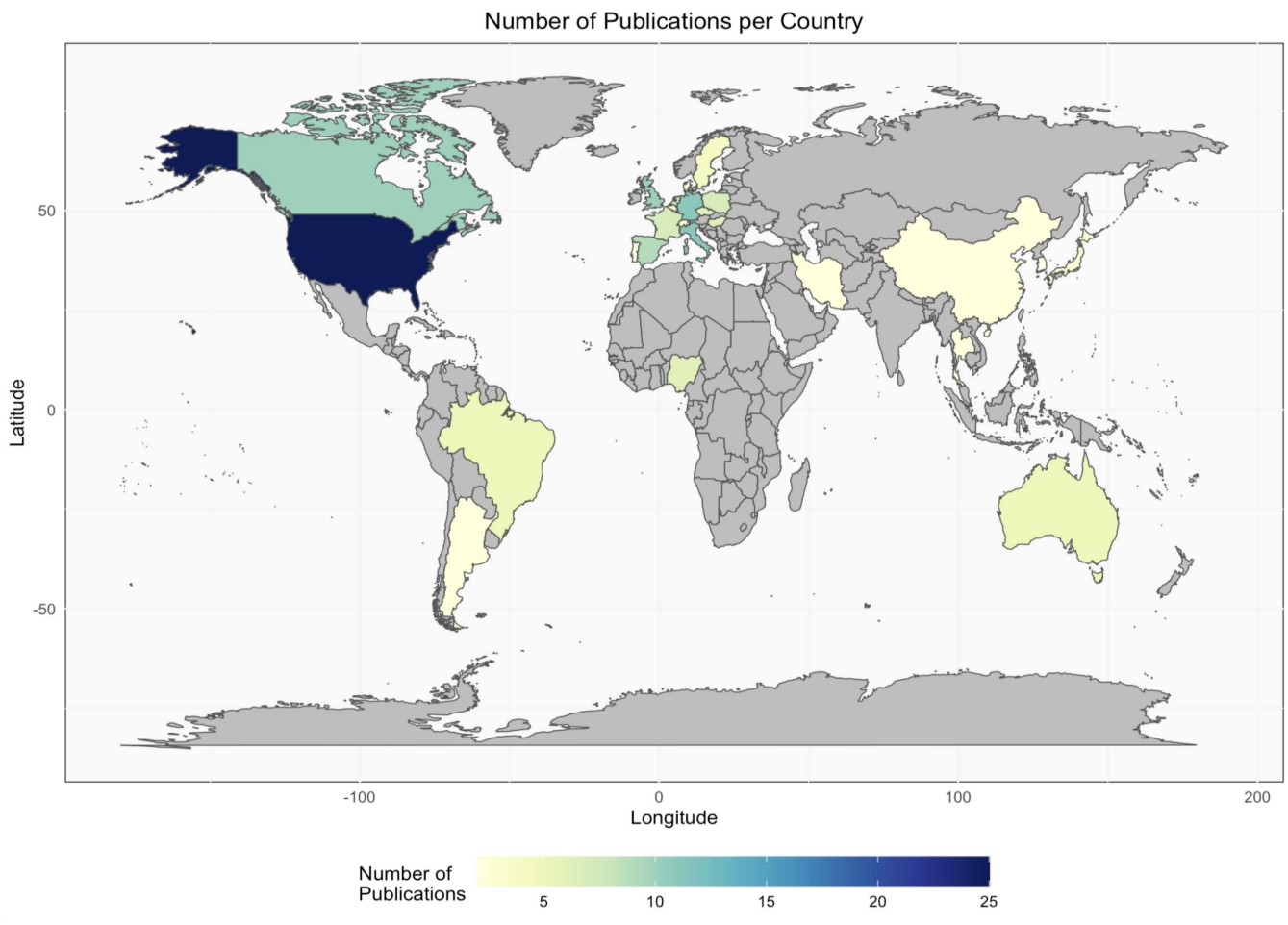

**Fig 2. A Choropleth map illustrating the geographical distribution and the quantity of research conducted on the antimicrobial resistance of *E. coli* in dogs and/or cats.**

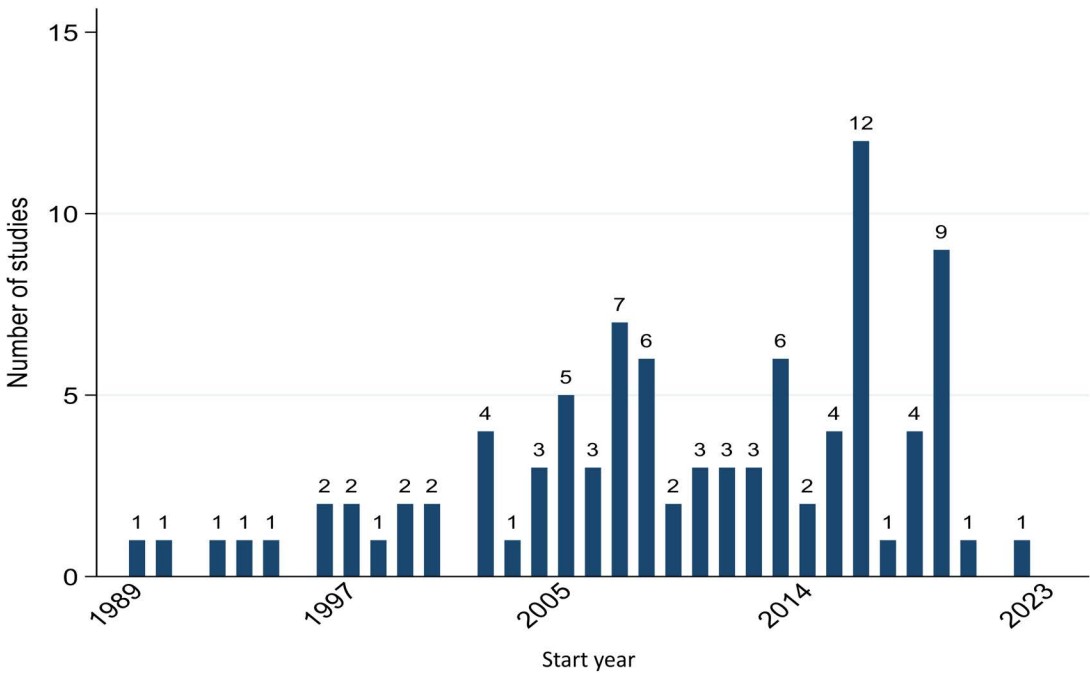

**Fig 3. A bar chart shows the number of studies on the antimicrobial resistance of *E. coli* in dogs and/or cats with the x-axis showing the initiation or the start year, and the y-axis showing the number of studies that were initiated in a specific year.**

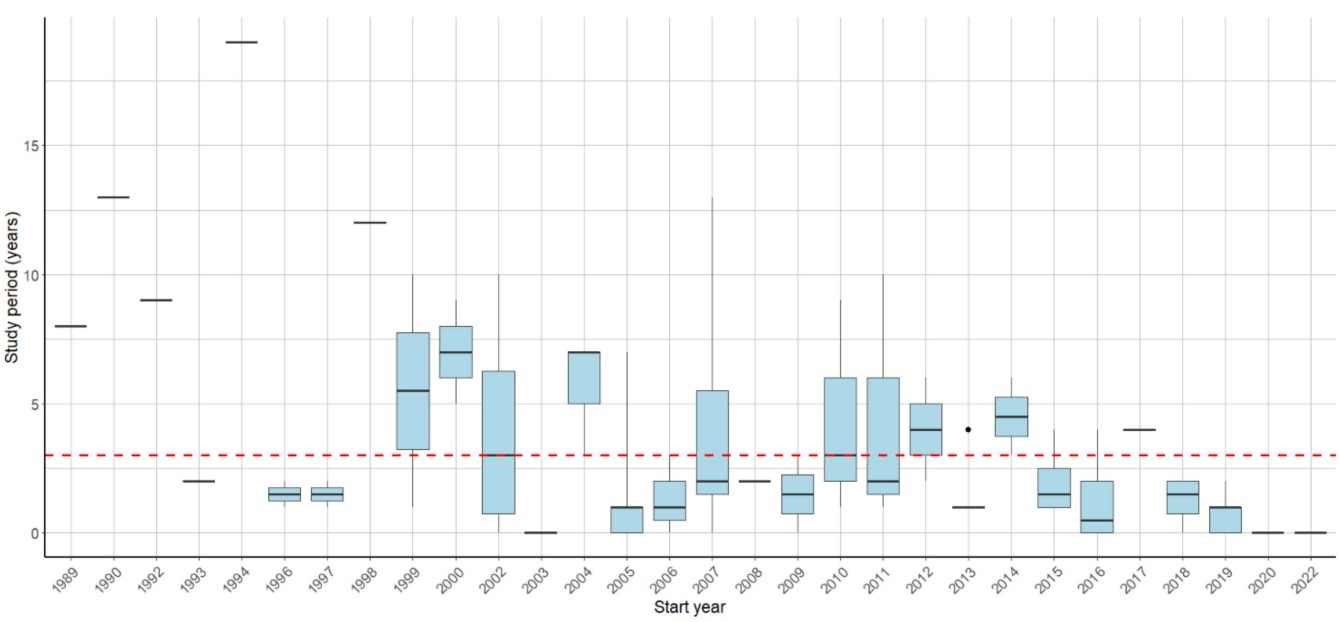

**Fig 4. A box plot showing the study period (initiation to completion year) of study(ies) that were conducted on the antimicrobial resistance of *E. coli* in dogs and/or cats.**

Of the included studies, 47.2% (n = 51) were conducted at the regional level (e.g., more than one facility), 40.7% (n = 44) were completed at an individual facility, 6.5% (n = 7) were international, and 5.6% (n = 6) were conducted at the national level. The European Animal Health Study Centre (CEESA) ComPath Study was undertaken exclusively at the international level (57.1%, n = 4) and was primarily an ongoing AMR monitoring and surveillance program for companion animals. About 66.7% (n = 72) of the studies included were conducted mainly at veterinary clinics (mostly primary care veterinary clinics), and these studies predominantly adopted an observational study design, as presented in **Table 1**. The study designs, as evaluated by the reviewers, were cross-tabulated with the study designs reported by the authors. Notably, 59.3% (n = 64) of the studies included did not report the study design (S4 Table).

About 97.2% (n = 105) of the included studies provided basic information on the source population of the pets enrolled. Regarding clinical history, 62.0% (n = 67) of the studies disclosed the clinical history of the pets enrolled. Of those with clinical history, 50.8% (n = 34) enrolled only sick pets, 35.8% (n = 24) enrolled only healthy pets, and 13.4% (n = 9) enrolled both sick and healthy pets. Regarding the recent antimicrobial treatment of pets whose clinical history was disclosed, 82.1% (n = 55) had no record of prior or recent antimicrobial therapy. In comparison, 17.9% (n = 12) had received antimicrobial treatment within the past 3 weeks, with 8 of the 12 studies reporting MDR *E. coli*, found in both healthy and sick pets. RCTs were exclusively conducted in studies that enrolled clinically healthy pets (12.5%, n = 3). These trials specifically designed to determine the effect of antimicrobials on the fecal microbiota of pets. *E. coli* was isolated from studies including sick pets (97.1%, n = 33) and healthy pets (70.8%, n = 17). *E. coli* and ESBL-*E. coli* were also isolated from studies including healthy pets (16.7%, n = 4) and sick pets (2.9%, n = 1). Urinary specimens were mainly submitted for microbiological analysis in studies that enrolled sick pets (32.4%, n = 11). In contrast, non-urinary samples were mainly submitted for microbial analysis in studies that enrolled healthy pets (100.0%, n = 24). In general, MDR *E. coli* was reported in studies that enrolled healthy and sick pets (Table 2). Overall, non-urinary specimens (54.6%, n = 59) were most commonly submitted for *E. coli* isolation in the studies included in this scoping review. *E. coli* and ESBL-*E. coli* were mostly isolated from studies that submitted non-urinary specimens (Table 3).

**Table 1. Summary of primary study settings and study design of included studies.**

| | Study Designs | | |
| --- | --- | --- | --- |
| | Observational | Antimicrobial monitoring and surveillance program | Randomized controlled trial |
| **Primary Study Settings**<br>**n (%)** | | | |
| Veterinary clinic | 65 (90.2) | 4 (5.5) | 3 (4.1) |
| Household | 10 (90.9) | 1 (9.1) | 0 (0.0) |
| Shelter | 5 (100.0) | 0 (0.0) | 0 (0.0) |
| Stray animals | 3 (100.0) | 0 (0.0) | 0 (0.0) |
| Veterinary clinic & household | 3 (100.0) | 0 (0.0) | 0 (0.0) |
| Shelter & household | 4 (100.0) | 0 (0.0) | 0 (0.0) |
| Veterinary clinic, shelter, & household | 1 (0.3) | 3 (0.7) | 0 (0.0) |
| Stray animals & household | 2 (100.0) | 0 (0.0) | 0 (0.0) |
| Veterinary clinic & stray animals | 1 (100.0) | 0 (0.0) | 0 (0.0) |
| Veterinary clinic & market | 1 (100.0) | 0 (0.0) | 0 (0.0) |
| Unknown source | 2 (100.0) | 0 (0.0) | 0 (0.0) |
| **Total no. of studies** | **97 (89.8)** | **8 (7.4)** | **3 (2.8)** |

n: number of studies.

**Table 2. Summary of study characteristics based on the clinical history of dogs and/or cats included in the studies.**

| Characteristic | Clinical history of pets | | |
| --- | --- | --- | --- |
| | Sick (n = 34) | Healthy (n = 24) | Sick & healthy (n = 9) |
| **Study Designs** | | | |
| Observational | 28 (82.4%) | 21 (87.5%) | 8 (88.9%) |
| Antimicrobial monitoring and surveillance program | 6 (17.6%) | 0 (0.0%) | 1 (11.1%) |
| Randomized controlled trial | 0 (0.0%) | 3 (12.5%) | 0 (0.0%) |
| **Bacterial isolated** | | | |
| *E. coli* | 33 (97.1%) | 17 (70.8%) | 8 (88.9%) |
| ESBL-*E. coli* | 0 (0.0%) | 3 (12.5%) | 0 (0.0%) |
| *E. coli* & ESBL-*E. coli* | 1 (2.9%) | 4 (16.7%) | 1 (11.1%) |
| **Specimen from which *E. coli* was isolated** | | | |
| Urinary | 11 (32.4%) | 0 (0.0%) | 2 (22.2%) |
| Non-urinary (e.g., feces) | 11 (32.4%) | 24 (100.0%) | 6 (66.7) |
| Urinary & non-urinary | 9 (26.5%) | 0 (0.0%) | 1 (11.1%) |
| Unknown | 3 (8.8%) | 0 (0.0%) | 0 (0.0%) |
| **Multi-drug resistance was reported in *E. coli*** | | | |
| Yes | 20 (58.8%) | 19 (79.2%) | 4 (44.4%) |
| No | 14 (41.2%) | 5 (20.8%) | 5 (55.6%) |
| **Consistency with standard MDR definition** | | | |
| Yes | 16 (80.0%) | 11 (57.9%) | 4 (100.0%) |
| No | 4 (20.0%) | 8 (42.1%) | 0 (0.0%) |
| **A carbapenem test was reported in *E. coli* or ESBL-*E. coli*** | | | |
| Yes | 8 (23.5%) | 10 (41.7%) | 5 (55.6%) |
| No | 26 (76.5%) | 14 (58.3%) | 4 (44.4%) |
| | | | |

n: number of studies.

**Table 3. Distribution of *E. coli* and extended-spectrum beta-lactamase-producing *E. coli* isolates from dogs and/or cats specimens submitted for microbiological analysis included in the studies.**

| Specimen | Bacteria isolated n (%) | | | |
| --- | --- | --- | --- | --- |
| | *E. coli* | ESBL-*E. coli* | *E. coli* & ESBL-*E. coli* | Total |
| Non-urinary | 50 (51.5) | 3 (100.0) | 6 (75.0) | 59 (54.6) |
| Urinary | 25 (25.8) | 0 (0.0) | 1 (12.5) | 26 (24.1) |
| Urinary and non-urinary | 16 (16.5) | 0 (0.0) | 1 (12.5) | 17 (15.7) |
| Unknown | 6 (6.2) | 0 (0.0) | 0 (0.0) | 6 (5.6) |
| Total | 97 (89.8) | 3 (2.8) | 8 (7.4) | 108 |

n: number of studies.

About 82.4% (n = 89) of the studies focused on determining the resistance rate of *E. coli* in dogs and cats at the isolate level, and 78.7% (n = 85) of the studies followed the Clinical and Laboratory Standard Institute (CLSI) guidelines throughout the designated study period. The average number of antimicrobials tested for susceptibility in *E. coli* isolates was 13, with a range from 1 to 35 (**Table 4**).

Disk diffusion (48.2%, n = 52) and broth microdilution (35.2%, n = 38) were the most common methods for AST in this study. All AMR monitoring and surveillance studies utilized the broth microdilution method for AST (100.0%, n = 8). Of the studies that exclusively used the broth microdilution method for AST, 84.2% (n = 32) reported the minimum inhibition concentrations (MICs) of the antimicrobials used. Regarding intermediate categorization, 38.9% (n = 42) of the studies did not provide information on the approach adopted for managing the intermediate category. Among those that did, the majority (33.3%, n = 36) reported it seperately, while some combined it with resistance data (19.4%, n = 21). Approximately 92.6% (n = 100) of the studies conducted AST on *E. coli* using the same panel of antimicrobials throughout the designated study period, while 7.4% (n = 8) did not (Table 6).

## Discussion

The result of this scoping review reveal the scope and depth of existing literature addressing the issue of AMR in *E. coli* among dogs and cats. This comprehensive exploration offersvaluable insights into the current state of AMR in companion animals. This research was primarily focused on the phenotypic dimensions of AMR. Including studies that emphasized the molecular basis of resistance could have broadened the scope beyond the predefined research questions and may not have directly contributed to addressing our specific objectives.

Over the past two decades, the increasing interest in AMR of companion animals was evident through a substantial rise in studies investigating AMR in *E. coli* from dogs and cats. While most studies had less than three years duration, a

**Table 4. Antimicrobial resistance rates, susceptibility guidelines, and the handling of intermediate categories for *E. coli* isolates from dogs and/or cats in the included studies.**

| The antimicrobial resistance rate in *E. coli* was based on: | n (%) |
|---|---|
| Isolates level | 89(82.4) |
| Animal level | 9 (8.3) |
| Animal and isolate level | 8 (7.4) |
| Unclear | 2 (1.9) |
| **Total number of studies** | **108** |
| **Antimicrobial susceptibility guidelines followed** | **n (%)** |
| CLSI | 85 (78.7) |
| CLSI & EUCAST | 11 (10.2) |
| Not reported | 5 (4.6) |
| Other | 4 (3.7) |
| Other & EUCAST | 2 (1.9) |
| CLSI, EUCAST, & Other | 1 (0.9) |
| **Total number of studies** | **108** |
| **How intermediate category was handled in the study** | **n (%)** |
| Reported separately | 36 (33.3) |
| Combined with resistance data | 21 (19.4) |
| Combined with susceptibility data | 9 (8.3) |
| Not reported | 42 (38.9) |
| **Total number of studies** | **108** |

EUCAST– The European Committee on Antimicrobial Susceptibility Testing

n – number of studies

A larger proportion of *E. coli* isolates tested for susceptibility originated from dogs (66.7%, n = 1,776) compared to cats (33.3%, n = 888) (**Table 5**).

**Table 5. Animal- and isolate-level antimicrobial susceptibility testing for *E. coli* from dogs and cats in the included studies.**

| Characteristic | Dogs | Cats | Total |
|---|---|---|---|
| The number of animals whose *E. coli* was tested for antimicrobial susceptibility at the animal level | 1,776 (66.7%) | 888 (33.3%) | 2,664 |
| The number of *E. coli* isolates tested for antimicrobial susceptibility at the isolate level | 423,570 (75.9%) | 134,320 (24.1%) | 557,890 |
| The number of animals and their corresponding *E. coli* isolates tested for antimicrobial susceptibility at the animal- and isolate-level | 956 (2,668) | 39 (195) | 995 (2,863) |

**Table 6. Methods of antimicrobial susceptibility testing and reporting of minimum inhibitory concentrations for *E. coli* isolates from dogs and/ or cats in the included studies.**

| Antimicrobial susceptibility testing method | n (%) | MIC reported n (%) | MIC not reported n (%) |
|---|---|---|---|
| Disk diffusion | 52 (48.2) | 4 (7.7) | 48 (92.3) |
| Broth microdilution | 38 (35.2) | 32 (84.2) | 6 (15.8) |
| Disk diffusion & broth microdilution | 6 (5.6) | 3 (50.0) | 3 (50.0) |
| Epsilometer test | 3 (2.8) | 3 (100.0) | 0 (0.0) |
| Broth microdilution & Epsilometer test | 1 (0.9) | 1 (100.0) | 0 (0.0) |
| Disk diffusion, broth microdilution, & Epsilometer test | 1 (0.9) | 1 (100.0) | 0 (0.0) |
| Disk diffusion & Epsilometer test | 1 (0.9) | 1 (100.0) | 0 (0.0) |
| Not reported | 6 (5.6) | 0 (0.0) | 6 (100.0) |
| **Total number of studies** | **108** | **45 (41.7)** | **63 (58.3)** |

considerable number exhibited extended durations, suggesting the potential to yield more comprehensive and valuable evidence. Recent investigations from diverse regions, including Europe, the United States, Canada, Australia, different African countries, and China, highlight the growing global concern regarding AMR in companion animals [18–22].

Given the significant role of veterinary clinics in providing of healthcare services to companion animals, researchers logically prioritized this setting over households and shelters. Veterinary clinics, being repositories of records for bacterial culture and AST, predominantly rely on historical data to conduct studies on AMR in companion animals. The adoption of this method, involving the analysis of existing datasets, is both pragmatic and justifiable when investigating patterns and trends associated with AMR profiles of bacteria affecting companion animals, as supported by Joosten et al. [23] and Singleton et al. [24]. Consequently, the choice of the observational study design was deemed reasonable for most studies included in this scoping review.

Samples for bacterial identification are predominantly collected from dogs and cats and subsequently sent to diagnostic laboratories, as described by Cummings et al. [25]. Within the scope of this study, urine emerged as the predominant clinical specimen collected from sick pets and submitted for diagnostic evaluation, likely due to its non-invasive nature and diagnostic relevance in providing comprehensive insights into the pathological conditions of these pets [26]. In contrast, non-urinary specimens, such as fecal samples, were more commonly collected from apparently healthy pets for *E. coli* isolation and AST, which may be attributed to the ease of collection and routine screening protocols [27]. These observations align with the findings of Salgado-Caxito et al. [8].

Veterinary practitioners frequently recommend AST for companion animal patients presenting with life-threatening conditionsor experiencing recurrent, nonresponsive, or chronic bacterial infections [28]. However, this review identified a limited number of studies that have undertaken RCTs to assess the comparative effectiveness of antimicrobial treatment options for companion animals, which can primarily be attributed to ethical concerns about exposing companion animals

to antimicrobials [29]. Additionally, practical challenges, such as the need for long-term follow-up, ensuring pet owners' compliance, and regulatory requirements, further limit the feasibility of RCTs in this context [30]. Furthermore, the integration of AST into AMR monitoring and surveillance programs for companion animals remains notably underexplored, likely due to resource constraints [18]. These findings underscore the urgent need to establish dedicated surveillance and monitoring programs to effectively address AMR in companion animals.

The exposure of companion animals to pathogenic and multidrug-resistant strains of *E. coli* is multifaceted, arising from various sources such as environmental factors, dietary intake; especially raw food consumption, and interactions with other animals [31]. The detection of *E. coli* and ESBL-*E. coli* from healthy pets does not necessarily indicate an active infection or disease [32]. Pets can harbor these bacteria asymptomatically, without manifesting any clinical signs [33]. However, this trend has raised growing concerns, particularly in recent studies from Europe and China, as noted by Cui et al. [34], Marco-Fuertes et al. [6], and Werhahn et al. [31]. Conversely, several studies have also identified MDR *E. coli* in sick dogs and cats. Fayez et al. [35] highlighted a higher prevalence of MDR and ESBL-*E. coli* in diseased cats, with recent antimicrobial treatment and cohabitation with humans as key risk factors. Similarly, Chen et al. [36] observed an increasing prevalence of ESBL-*E. coli* in sick cats and dogs in China, correlating this rise with heightened antimicrobial usage. These findings emphasize the need to address MDR *E. coli* in companion animals, regardless of their clinical status, to mitigate the risk of potential transmission to humans. The implementation of stringent hygiene practices and adherence to antimicrobial stewardship principles in veterinary medicine is crucial to preventing the spread of MDR *E. coli* between pets and their owners, as emphasized by Guardabassi et al. [37].

Antimicrobial susceptibility examination using a diverse array of antimicrobials plays a crucial role in comprehending the resistance patterns of bacterial isolates and guiding the selection of appropriate antimicrobial therapies [38]. In this review, most studies assessed *E. coli* using a consistent panel of antimicrobials throughout the study period, indicating a standardized approach to AST. Such consistency enhances the reliability of comparisons across time and populations, supports trend analysis, and contributes to more robust surveillance data. However, the number of antimicrobials included in the panel for AST may differ, both within and across study settings, contingent upon factors such as the research objective, local resistance patterns, and the clinical relevance of the antimicrobials selected for evaluation, as explained by Finegold [39] and Desmet et al. [40]. *E. coli* was isolated and subjected to AST more frequently among dogs than in cats, a discrepancy that may be attributed to a higher prevalence of sampling within the dog population. Similar investigations have reported this notable difference [41,42].

The use of disk diffusion and broth microdilution methods in veterinary diagnostic practices is well-established, with studies demonstrating its effectiveness in determining antimicrobial susceptibility for various bacterial pathogens affecting companion animals [43,44]. Broth microdilution has emerged as the preferred method for AMR monitoring and surveillance programs targeting *E. coli* in companion animals, consistent with the findings of Pedersen et al. [45] and Thungrat et al. [46]. The advantages of the broth microdilution assay include its precision, reproducibility, time efficiency, cost-effectiveness, and ability to concurrently assess multiple antimicrobials [47,48]. This method allows for the determination of MICs for antimicrobials, which is a critical aspect in AMR monitoring and surveillance efforts [47]. Furthermore, broth microdilution is recognized as a reference method recommended by the CLSI and EUCAST for AMR monitoring and surveillance [48]. Both CLSI and EUCAST guidelines hold significant global influence on AST standards in clinical microbiology laboratories, with EUCAST being predominant in Europe [49]. These guidelines establish breakpoints; concentrations of antimicrobial agents that categorize bacterial isolates as susceptible, intermediate, or resistant in AST. In cases where CLSI lacks established breakpoints, laboratories may adopt EUCAST guidelines as an alternative. Studies have shown that transitioning from CLSI to EUCAST standards may alter antimicrobial resistance profiles in both clinical and commensal *E. coli* [50]. The impact of these changes on empirical treatment protocols varies based on context, and such variations can significantly influence susceptibility interpretation and AMR surveillance outcomes [51].

This scoping review highlighted several key knowledge gaps that could inform future research. A notable observation is the predominant focus of studies on the AMR of *E. coli* in dogs and cats within veterinary clinics. This raises concerns about the representativeness of research findings, particularly in low and middle-income countries, where factors such as limited access to veterinary care may hinder the inclusivity of data. This limitation may result in the underrepresentation of stray or free-roaming household pets, as well as those owned by lower socioeconomic owners, a substantial portion of the dog and cat population both locally and globally . Limited expansion of study settings beyond veterinary clinics represents a critical gap which should be addressed in future studies. Additionally, the choice of the unit of analysis for resistance rate (e.g., animal-level, isolate-level, or both) can significantly impact the reported AMR rates. This variability in the unit of analysis introduces challenges in making direct comparisons across studies. Standardized reporting practices and transparent communication of methods and results are crucial for comprehending and synthesizing data from diverse sources. It is crucial to provide details regarding the duration of AMR studies, as this information is essential for identifying temporal variations in AMR across different studies, thereby enhancing our understanding of the dynamics of AMR in companion animals. The interpretation and reporting of intermediate results varied among studies. While some studies combined intermediate with susceptibility data, others integrated it with resistance data, introducing complexities in data interpretation and comparison. Comparing AMR studies proves challenging due to disparities in result analysis, particularly regarding handling the intermediates category. The need for standardized approaches across studies enables direct and comprehensive comparisons of findings, facilitating a more accurate assessment of AMR trends and contributing to a unified understanding of the issue. Reporting intermediate susceptibility separately provides more precise information regardingthe level of resistance. It allows for a more nuanced interpretation, valuable in research and surveillance studies, offering insights into the evolving landscape of AMR in companion animals. Furthermore, certain studies reported MDR in *E. coli* but deviated from the standard definition of MDR, which is characterized by resistance to one or more drugs in three or more classes, a definition widely accepted in both human and veterinary medicine [52,53]. Notably, many authors of the studies included in this scoping review omitted reporting the study design and this lackof information raises concerns in scientific research, as it hinders transparency, reproducibility, and the critical evaluation of study findings [54]. A clear and accurate description of the study design is essential for understanding the methodology, potential biases, and the strength of evidence provided by the research [54]. Different study designs have distinct strengths and limitations, making knowledge of the design essential for appropriately interpreting results. The reviewers inferred the study design based on the available information in the articles, although this information was not always sufficiently detailed.

## Study limitations

While the study aimed for rigor and coherence in the selection process, certain limitations were acknowledged. The exclusion of non-primary research and descriptive studies may have led to the omission of potentially relevant information, limiting the comprehensive nature of the scoping review. These restrictive criteria could result in an underrepresentation of certain perspectives or data sources that might contribute valuable insights to this study. Additionally, the language restriction to English only could introduce selection bias by excluding potentially relevant studies not written in English, thereby limiting the global generalizability of the findings.

## Conclusion

Studies on the antimicrobial resistance of *E. coli* in companion animals are largely concentrated in North America and Western Europe, with antimicrobial susceptibility testing guiding treatment decisions in veterinary clinics.The detection of multidrug-resistant *E. coli* in healthy and sick pets calls for antimicrobial stewardship, yet there are limited surveillance programs on this rising issue.

## Supporting information

**S1 Checklist. PRISMA-P checklist.**
(PDF)

**S2 Checklist. PRISMA-ScR checklist.**
(PDF)

**S1 Appendix. S1 AppendixA. Post-protocol deviation** .
(PDF)

**S2 Appendix. S2 AppendixB. Data items.**
(PDF)

**S1 Table. Summary and data charting of articles: excluded articles with reasons, eligible articles, and charted data from included studies.**
(PDF)

**S2 Table. Frequency of conflicts and the corresponding Kappa coefficient at each of the screening and data charting stages of the study selection process.**
(PDF)

**S3 Table. Affiliations of the first authors involved in research on the antimicrobial resistance of *E. coli* in dogs and/or cats.**
(PDF)

**S4 Table. Cross-tabulation of the study designs as evaluated by the reviewers with study designs reported by the authors of the research on the antimicrobial resistance of *E. coli* in dogs and/or cats.**
(PDF)

## Acknowledgments

The development of the search strategy for this scoping review benefitted significantly from the valuable assistance of Nancy Birch, Associate Librarian at the University of Guelph. We extend our sincere gratitude to her for her significant support on this work.

## Author contributions

**Conceptualization:** Rasaq A. Ojasanya, Zvonimir Poljak, Kurtis E. Sobkowich, Theresa M. Bernardo, J. Scott Weese.

**Data curation:** Rasaq A. Ojasanya, Kurtis E. Sobkowich, Uththami Kukathasan.

**Formal analysis:** Rasaq A. Ojasanya.

**Funding acquisition:** Theresa M. Bernardo.

**Investigation:** Rasaq A. Ojasanya, J. Scott Weese, Zvonimir Poljak, Uththami Kukathasan, Kurtis E. Sobkowich.

**Methodology:** Rasaq A. Ojasanya, Zvonimir Poljak, Kurtis E. Sobkowich, Theresa M. Bernardo, J. Scott Weese.

**Project administration:** Zvonimir Poljak, Kurtis E. Sobkowich, Theresa M. Bernardo.

**Resources:** Theresa M. Bernardo.

**Software:** Zvonimir Poljak, Kurtis E. Sobkowich, Theresa M. Bernardo.

**Supervision:** J. Scott Weese, Zvonimir Poljak, Theresa M. Bernardo.

**Validation:** Rasaq A. Ojasanya, J. Scott Weese, Zvonimir Poljak, Kurtis E. Sobkowich, Uththami Kukathasan, Theresa M. Bernardo.

**Visualization:** Rasaq A. Ojasanya, Kurtis E. Sobkowich.

**Writing – original draft:** Rasaq A. Ojasanya.

**Writing – review & editing:** Rasaq A. Ojasanya, J. Scott Weese, Zvonimir Poljak, Kurtis E. Sobkowich, Theresa M. Bernardo.

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
