## [Decision Letter · Decision Letter 0]

13 Dec 2024

PONE-D-24-06231Antimicrobial-resistance of Escherichia coli in dogs and cats: A scoping reviewPLOS ONE

Dear Dr. Ojasanya,

Thank you for submitting your manuscript to PLOS ONE. After careful consideration, we feel that it has merit but does not fully meet PLOS ONE’s publication criteria as it currently stands. Therefore, we invite you to submit a revised version of the manuscript that addresses the points raised during the review process.

**ACADEMIC EDITOR:****Revise the manuscript as per comments from the reviewers.**==============================

We look forward to receiving your revised manuscript.

Kind regards,

Naveed Ahmed, Ph.D

Academic Editor

PLOS ONE

2. Thank you for stating the following financial disclosure: [The authors would like to thank IDEXX Laboratories for their support of the IDEXX Chair in Emerging Technologies and Preventive Healthcare, which funded this research.].  Please state what role the funders took in the study.  If the funders had no role, please state: "The funders had no role in study design, data collection and analysis, decision to publish, or preparation of the manuscript." If this statement is not correct you must amend it as needed. Please include this amended Role of Funder statement in your cover letter; we will change the online submission form on your behalf.

3. As required by our policy on Data Availability, please ensure your manuscript or supplementary information includes the following:

4. We note that Figure 2 in your submission contain [map/satellite] images which may be copyrighted. All PLOS content is published under the Creative Commons Attribution License (CC BY 4.0), which means that the manuscript, images, and Supporting Information files will be freely available online, and any third party is permitted to access, download, copy, distribute, and use these materials in any way, even commercially, with proper attribution. For these reasons, we cannot publish previously copyrighted maps or satellite images created using proprietary data, such as Google software (Google Maps, Street View, and Earth). For more information, see our copyright guidelines: http://journals.plos.org/plosone/s/licenses-and-copyright.

1. You may seek permission from the original copyright holder of Figure 2 to publish the content specifically under the CC BY 4.0 license.  

Additional Editor Comments:

Revise the manuscript as per comments.

Reviewers' comments:

Reviewer's Responses to Questions

**Comments to the Author**

1. Is the manuscript technically sound, and do the data support the conclusions?

Reviewer #1: Yes

Reviewer #2: Yes

2. Has the statistical analysis been performed appropriately and rigorously? 

Reviewer #1: Yes

Reviewer #2: Not Applicable.

3. Have the authors made all data underlying the findings in their manuscript fully available?

Reviewer #1: Yes

Reviewer #2: Yes

4. Is the manuscript presented in an intelligible fashion and written in standard English?

Reviewer #1: Yes

Reviewer #2: Yes

5. Review Comments to the Author

Reviewer #1: Figures 1,2,3 and 4 are not visible. Those must be incorporated accordingly. Data in Tables is shown in some tables as n(%) while in Table-4 n(%) is mentioned in the title. All tables must be symetric.

Reviewer #2:

I have reviewed the manuscript entitled “Antimicrobial-resistance of Escherichia coli in dogs and cats: A scoping review” submitted for possible publication in the journal “PlosOne”. The manuscript highlights a very important topic which is AMR. The authors have done great efforts in compiling all of the relevant literature in the manuscript, However, it needs some correction before it can be considered for further processing. My specific comments are:

Because the study is more towards systematic search, I would like to recommend to put “Systematic and a scoping review” at the end of article title.In the search strategy, the authors should use “*Enterobacteriaceae* ; *Enterobacterales* ” in order to make sure that there is no article left for *E. coli* .At the end of introduction section, the authors should provide the sudy rationale and justification. What is the need to conduct study on *E. coli* only? There are plenty of other organism related to the animals.In the articles included, did the authors found any information regarding the type of cats and dogs, which means whether they were pets or street animals?The use of abbreviations is appropriate. The authors should carefully check again the use of abbreviations throughout the manuscript. for example, at line 50 and 51 the word “antimicrobial resistance” is used twice but the abbreviation was mentioned on the second time while any abbreviation should be mentioned at it’s first appearance.The authors should carefully check the organism’s name and follow the same throughout the manuscript. For example, at some place they mentioned full form of *Escherichia coli* , and at certain levels only *E. coli* .Line 52-53: The authors should mention the background only related to the companion animals. Remove the irrelevant background.Line 73: Mention about how the protocol was prepared.Line 88-89: Remove the sentence “this review has one…..”In eligibility criteria, what type of studies were considered for this article? I mean whether any case series etc.?Table 1: This table is not necessary and can be removed. Also, the number of studies not matching with the final studies included in the article.Table 3, 4: replace the legend with “n: number of studies”.Table 5 is very confusing. The authors are recommended to redesign the table and split in 3 or 4. Some of the information can be described in the text.

6. PLOS authors have the option to publish the peer review history of their article (what does this mean? ). If published, this will include your full peer review and any attached files.

**Do you want your identity to be public for this peer review?** For information about this choice, including consent withdrawal, please see our Privacy Policy .

Reviewer #1: No

Reviewer #2: Naveed Ahmed

---

## [Author Response · Author response to Decision Letter 1]

24 Jan 2025

The response to reviewers has been provided in the letter titled, "Response to Reviewers".

---

## [Editor Report · Decision Letter 1]

5 Feb 2025

PONE-D-24-06231R1Antimicrobial-resistance of Escherichia coli in dogs and cats: a scoping reviewPLOS ONE

Dear Dr. Ojasanya,

Thank you for submitting your manuscript to PLOS ONE. After careful consideration, we feel that it has merit but does not fully meet PLOS ONE’s publication criteria as it currently stands. Therefore, we invite you to submit a revised version of the manuscript that addresses the points raised during the review process.

**ACADEMIC EDITOR:**Please revise the manuscript as per the comments given in "Additional Editor Comments"==============================

We look forward to receiving your revised manuscript.

Kind regards,

Naveed Ahmed, Ph.D

Academic Editor

PLOS ONE

Journal Requirements:

Additional Editor Comments:

1. Table 1 Line 116: needs to be removed and replace with the text (a paragraph).

2. Figure 3: Mention the data numbers on each bar.

3. Figure 1 and 2: The resolution need to increased.

4. Figure 4: The error bars needs to be consistent. Either put the error bars of all, or else remove it.

5. Table 1 Line 271, Table 2 and Table 3: The table legends needs to be set.

---

## [Author Response · Author response to Decision Letter 2]

13 Mar 2025

The shapefile of the world was retrieved from the Database of Global Administrative Areas (https://gadm.org/download_country.html), imported into R (R Core Team, 2023), and used to represent the distribution of retrieved publications on the map. This has been cited in the manuscript with track changes (lines 177–179), with the reference provided as number 17.

---

## [Editor Report · Decision Letter 2]

7 Apr 2025

Antimicrobial-resistance of Escherichia coli in dogs and cats: a scoping review

PONE-D-24-06231R2

Dear Dr. Ojasanya,

We’re pleased to inform you that your manuscript has been judged scientifically suitable for publication and will be formally accepted for publication once it meets all outstanding technical requirements.

Kind regards,

Naveed Ahmed, Ph.D

Academic Editor

PLOS ONE
---

## [Editor Report · Acceptance letter]

PONE-D-24-06231R2

PLOS ONE

Dear Dr. Ojasanya,

I'm pleased to inform you that your manuscript has been deemed suitable for publication in PLOS ONE. Congratulations! Your manuscript is now being handed over to our production team.

Kind regards,

on behalf of

Dr. Naveed Ahmed

Academic Editor

PLOS ONE